# Rhegmatogenous Retinal Detachment in Musculocontractural Ehlers–Danlos Syndrome Caused by Biallelic Loss-of-Function Variants of Gene for Dermatan Sulfate Epimerase

**DOI:** 10.3390/jcm12051728

**Published:** 2023-02-21

**Authors:** Yuji Yoshikawa, Takashi Koto, Tomoka Ishida, Tomoko Uehara, Mamiko Yamada, Kenjiro Kosaki, Makoto Inoue

**Affiliations:** 1Kyorin Eye Center, School of Medicine, Kyorin University, 6-20-2 Shinkawa, Mitaka 181-8611, Tokyo, Japan; 2Department of Ophthalmology, Faculty of Medicine, Saitama Medical University, 38 Morohongo, Moroyama, Iruma 350-0495, Saitama, Japan; 3Center for Medical Genetics, School of Medicine, Keio University, 35 Shinanomachi, Shinjyuku, Tokyo 160-8582, Tokyo, Japan

**Keywords:** musculocontractural Ehlers–Danlos syndrome, dermatan sulfate epimerase, rhegmatogenous retinal detachment

## Abstract

Musculocontractural Ehlers–Danlos syndrome, caused by biallelic loss-of-function variants for dermatan sulfate epimerase (mcEDS-DSE), is a rare connective tissue disorder. Eight patients with mcEDS-DSE have been described with ocular complications, including blue sclera, strabismus, high refractive errors, and elevated intraocular pressure. However, a case with rhegmatogenous retinal detachment (RRD) has not been reported. We report our findings in a 24-year-old woman who was diagnosed with mcEDS-DSE in childhood and presented to our clinic with an RRD in the left eye. The RRD extended to the macula and was associated with an atrophic hole. The patient underwent scleral buckling surgery and cryopexy with drainage of subretinal fluid through a sclerotomy under local anesthesia. The sclera did not appear blue but was very thin at the sclerotomy site. The patient developed frequent bradycardia during the surgery. Subretinal or choroidal hemorrhages were not observed intraoperatively; however, a peripapillary hemorrhage was observed one day after operation. The retina was reattached postoperatively, and the peripapillary hemorrhage was absorbed after one month. The peripapillary retinal hemorrhages, thin sclera, and bradycardia were most likely due to the fragility of the eye. The genetic diagnosis of mcEDS-DSE played an important role before and during the surgery by alerting the surgeons to possible surgical complications due to the thin sclera.

## 1. Introduction

Recent advances in genetic diagnosis have increased the number of cases in which a genetic diagnosis is possible, even for rare diseases. It is expected that the ability to predict the phenotype will identify possible complications that might arise during surgery.

Ehlers–Danlos syndrome (EDS) is an inherited connective tissue disorder characterized by hyperextensibility of the skin, hypermobility, and tissue fragility [1]. Musculocontractural Ehlers–Danlos syndrome (mcEDS) is a subtype of EDS that is due to a deficiency in systemic dermatan sulfate [1]. This deficiency results from defective biosynthesis of dermatan sulfate caused by a biallelic loss-of-function variant of carbohydrate sulfotransferase 14 (CHST14, mcEDS-*CHST14*) or dermatan sulfate epimerase (DSE, mcEDS-DSE) [2,3].

Patients with mcEDS are characterized by developmental abnormalities, e.g., congenital contractures of multiple joints, and abnormalities of the cardiovascular system, respiratory system, urinary system, central nervous system, and ocular systems. There is a progressive increase in the fragility of the connective tissue, resulting in hyperextension and skin fragility, generalized joint laxity, recurrent joint dislocation, skeletal deformities, and giant subcutaneous hematomas [2,3]. mcEDS-DSE is a congenital multisystem disorder with symptoms similar to mcEDS-*CHST14*; however, the severity of the symptoms is lower in mcEDS-DSE [3]. Eight mcEDS-DSE patients, including three men and five women (two Indian, two Spanish, one Portuguese, one Pakistani, one Turkish, and one unknown) have been described in the literature with ocular complications, including blue sclera, strabismus, high refractive error, and elevated intraocular pressure [3,4,5]. These eight patients with mcEDS-DSE were reported to have craniofacial hypertelorism in six of six cases (100%), down-slanting palpebral fissures in seven of eight (88%), ear deformity in seven of eight (88%), congenital multiple contractures in seven of eight (88%), talipes equinovarus in six of eight (75%), long or slender finger shape characteristics in eight of eight (100%), progressive foot deformities in six of six (100%), skin hyperextensibility in three of seven (43%), fine or acrogeria-like palmar creases in seven of eight (88%), and large subcutaneous hematoma in four of five (80%) [5]. However, no complications of retinal detachment (RD) have been reported. By contrast, 6 of the 41 patients with mcEDS-CHST14 were reported to have RD [3].

We report our findings in a young woman with genetically diagnosed mcEDS-DSE who developed a unilateral RRD that was treated successfully with scleral buckling surgery. The surgery was performed with prior knowledge of the possible complications associated with the tissue abnormalities in the eyes in mcEDS-DSE.

## 2. Case Report

A 24-year-old young woman noticed a visual field defect in her left eye. She had been diagnosed with mcEDS in her childhood at Keio University Hospital. Genetic analysis found that mcEDS was caused by a biallelic loss-of-function variant of DSE. An exome analysis revealed that she had a compound heterozygous missense and frameshift mutations in exon 6 of the *DSE* gene: c. 1633T>G, p. Tyr545Asp, and c.1160delG, p.Gly387Alafs*14. She was diagnosed with unilateral RRD without macular involvement at a local hospital but declined surgery because of the many known complications associated with mcEDS.

Then, she visited the Kyorin Eye Center, and our examination showed that her best corrected visual acuity (BCVA) in the left eye was 20/33, the refractive error (spherical equivalent) was −9.0 diopters (D), and the axial length was 27.7 mm. Slit lamp examinations did not show a blue sclera (Figure 1), and the cornea, anterior chamber, lens, and vitreous body were clear.

Fundus examinations detected an RRD in the nasal and inferior quadrants that was located near the macula and was associated with an atrophic hole in the inferonasal quadrants and lattice degeneration in the temporal quadrants. The posterior vitreous was not detached. Fundus autofluorescence (FAF) examinations showed a hypo-FAF area corresponding to the RRD and hyper-FAF lesions corresponding to the subretinal fibrosis. However, we did not observe peripapillary starfish hypo-FAF caused by a fissure in Bruch’s membrane, which is observed in eyes with retinal angioid streaks and which is one of the subtypes of EDS (Figure 2). The right eye appeared to be normal except for a myopic change in the fundus.

Because of the fragility of the eye due to mcEDS, minimal surgery was planned with minimal deformation of the eyeball. Before the scleral buckling surgery, retinal photocoagulation was performed for the lattice degeneration at the temporal quadrant without an RRD to reduce the area needing the scleral buckle. Then, the scleral buckling surgery was performed for the RRD under local anesthesia. The surgery was begun by placing 3.0 tractional silk sutures on the inferior and internal rectus muscles to manage the position of the eyeball. During the placement of the tractional sutures, prominent bradycardia was observed, so it was necessary to proceed with the surgery while reducing the traction on the eyeball. Retinal cryopexy was performed around the atrophic holes, and a #506 silicone sponge buckle was sutured between 6 and 11 o’clock. A scleral incision was made at 8 o’clock to drain the subretinal fluid, and a very thin sclera was found at the sclerotomy site (Figure 3).

The site of the sclerotomy was sufficiently coagulated, and then the uvea was perforated with a fine needle. The thick subretinal fluid was drained without depressing the sclera to avoid deformation of the eyeball and causing subretinal hemorrhage due to the fragility of the sclera and Bruch’s membrane. We did not find any subretinal or choroidal hemorrhages, including at the sclerotomy site during the operation, and the surgery was completed after confirming that the subretinal fluid was reduced and the atrophic hole was positioned beneath the scleral buckle.

On the day after the surgery, the atrophic hole was found to be beneath the buckle in a proper position, and the subretinal fluid was reduced, but a peripapillary subretinal hemorrhage that extended to the upper and lower arcade vessels was observed (Figure 4). The postoperative FAF image showed a hypo-FAF area corresponding to the peripapillary hemorrhage. The peripapillary hemorrhage disappeared one month after the surgery and the FAF image showed no peripapillary starfish hypo-FAF lesion suggestive of Bruch’s membrane or RPE cracks, as was observed preoperatively.

## 3. Discussion

EDS is a systemic disorder that causes degeneration of the ocular connective tissue and a thinning of the retina due to scleral extensions and vitreous degeneration [6]. These abnormalities are risk factors for retinal detachments [7]. Previous studies have reported cases of EDS with RD (Table 1) [8,9,10,11,12,13,14].

Minatogawa and associates [15] reviewed 66 patients with mcEDS-*CHST14*, including 48 patients who had been reported earlier. The ophthalmologic complications were refractive errors including myopia, astigmatism, hyperopia, and amblyopia. As opposed to eyes with mcEDS-DSE, RD was present in six patients whose median age was 15.5 years with a range of 9 to 20 years [10,11,12,15] (Table 1). Retinal angioid streaks were present in two patients [15]. Blindness was present in six patients, and it was associated with glaucoma, macular bleeding, lacquer crack in Bruch’s membrane, and RRD [15].

The *CHST14* gene encodes N-acetylgalactosamine 4-O-sulfotransferase 1 (D4ST1), which catalyzes the 4-O sulfation of N-acetylgalactosamine and is involved in the biosynthesis of dermatan sulfate. Dermatan sulfate plays an important role in human development and extracellular-matrix maintenance [16]. Abnormalities in dermatan acid result in severe systemic complications, but mutations in *DSE* encoding dermatan sulfate epimerase-1 (DS-epi1) partially maintain the dermatan sulfate activity [17]. Thus, no severe systemic complications have been reported compared with the *CHST14* mutations [3].

Our case of mcEDS-DSE had RRD that was successfully reattached with scleral buckling surgery with precautions taken because of the fragility of the eye. The patient did not have vitreous degeneration and blue sclera, but the RRD in the left eye was most likely due to high myopia, which is a risk factor for RRD.

The intraoperative observations showed that the sclera was very thin and required careful manipulations, and the heart monitor needed to be continuously watched because the manipulation of the ocular tissue was associated with bradycardia. Creating the scleral incision for subretinal fluid drainage and placing sutures for the scleral buckling also required careful manipulations. Intraoperatively, no subretinal and choroidal hemorrhages were detected, but intraretinal and subretinal hemorrhages around the optic disc extending to the upper and lower arcade vessels were detected postoperatively. FAF examinations showed no peripapillary starfish hypo-FAF lesions pre- and postoperatively which would have indicated the presence of cracks in Bruch’s membrane or RPE layers as with retinal angioid streaks. These findings suggest that the subretinal hemorrhage may have been from peripapillary vessels due to the scleral fragility and vascular fragility [18,19]. The placement of the scleral buckle may have stressed the area around the optic nerve where the shape of the eye changes the most, causing the breakage of peripapillary vessels and leading to hemorrhage. The intraoperative surgical procedures or a rupture of the Bruch’s membrane, as is seen in simple hemorrhages in eyes with high myopia [20], did not appear to be the cause of the peripapillary subretinal hemorrhage according to the intraoperative and postoperative findings.

A previous report of vitrectomy for RRD in type VI EDS reported significant intraoperative hemorrhages and difficulty in closing the scleral wound due to a thinning of the sclera (Table 1) [9]. The vascular fragility in EDS is due to the fragility of the perivascular connective tissue, and it is especially prominent in type IV (vascular) EDS, which is caused by a molecular defect in type Ⅲ collagen. However, the other subtypes are also known to have varying degrees of vascular fragility [18,19]. Although the triggers for the hemorrhages are not clear, it is possible that acute changes in the intraocular pressure due to drainage or ocular deformations with traction on the eye during the surgical manipulation may have contributed to the bleeding. Severe intraoperative bradycardia due to the oculocardiac reflex in response to traction of the extraocular muscles was also prominently observed in our patient. Previous studies have reported autonomic failure due to the presence of cardiovascular dysfunction in EDS, and intraoperative systemic management should be carefully monitored [21].

In earlier studies of RDs in EDS patients, vitrectomy was used to treat RD because of the presence of scleral atrophy and thinning [9,14]. In another case, vitrectomy with a 360-degree scleral graft was used because of the extensive scleral ectasia and sclera ruptures during the reattachment of the RD in EDS patients [13]. In these cases, the subtype of EDS was not reported. However, in our case, although intraoperative scleral thinning and postoperative retinal hemorrhage were observed, scleral sutures of silicone sponge and scleral incision could be performed as usual while expecting possible complications from the fragility of the eyeball. Our case had mcEDS-DSE with an EDS mutation, and the maintenance of the partial activity of dermatan sulfate preserved the architecture of the ocular tissues. This allowed the scleral buckling surgery to be performed, unlike in eyes with conventional EDS. Thus, identifying the causative gene and phenotypes will provide important information for surgical strategies.

Our study has several limitations. First, this was a single case study, and statistical analysis could not be performed. Nevertheless, scleral buckling surgery was performed successfully. Second, the eye was highly myopic, which is a risk factor for developing RRD and one of the eye complications in mcEDS. Thus, further studies on a larger number of mcEDS-DSE patients are needed.

## 4. Conclusions

We conclude that cases of mcEDS-DSE with RRD can be successfully treated by scleral buckling surgery. We also conclude that knowing the genetic diagnosis will provide helpful information on possible complications during the intra- and postoperative periods. We recommend that a preoperative genetic diagnosis be made in all cases of inherited connective tissue disorders and that the surgical procedures be designed appropriately.

## Figures and Tables

**Figure 1 jcm-12-01728-f001:**
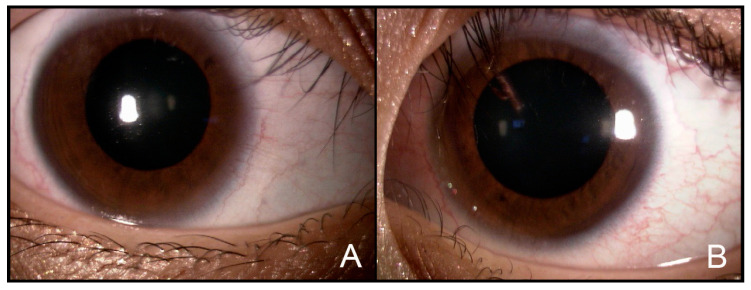
Photograph of the anterior segment of the right and left eyes. There is no evidence of a blue sclera, corneal opacities, or cataracts in either eye ((**A**): right eye, (**B**): left eye).

**Figure 2 jcm-12-01728-f002:**
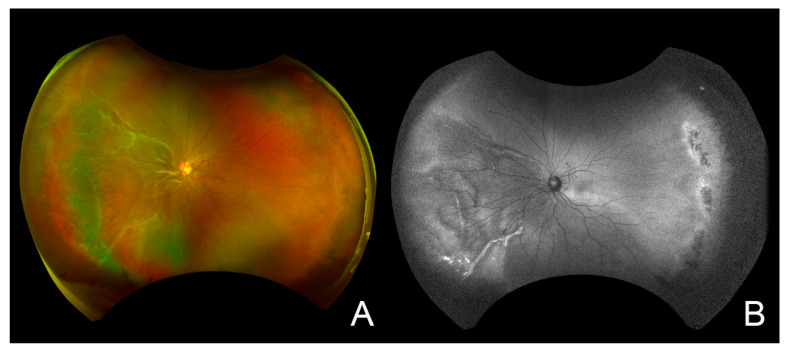
Preoperative widefield fundus photograph and fundus autofluorescence (FAF) image of the left eye. (**A**): The widefield fundus photograph shows the retinal detachment in the nasal and inferior quadrants adjacent to the macula. It is associated with an atrophic hole in the inferonasal quadrants and lattice degeneration in the temporal quadrants. (**B**): FAF image shows a hypo-FAF area corresponding to retinal detachment and hyper-FAF lesions corresponding to the subretinal fibrosis, but no peripapillary starfish hypo-FAF was observed in the eye with retinal angioid streaks.

**Figure 3 jcm-12-01728-f003:**
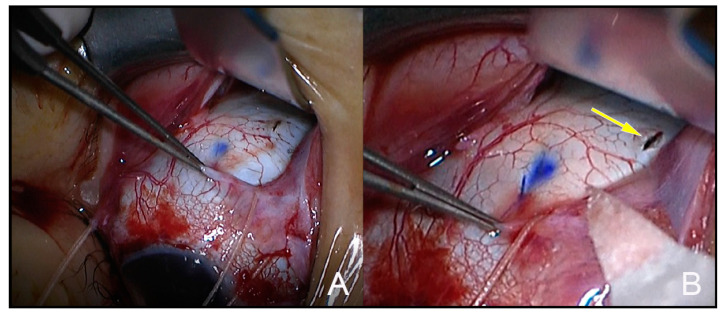
Intraoperative photograph of the inferonasal sclera during sclerotomy. (**A**): Scleral incision was made inferior to the internal rectus muscle to drain the subretinal fluid. Blue sclera was not observed. (**B**): Magnified image shows that the sclera at the site of sclerotomy (yellow arrow) was very thin. No choroidal hemorrhage could be seen at the sclerotomy site during the drainage.

**Figure 4 jcm-12-01728-f004:**
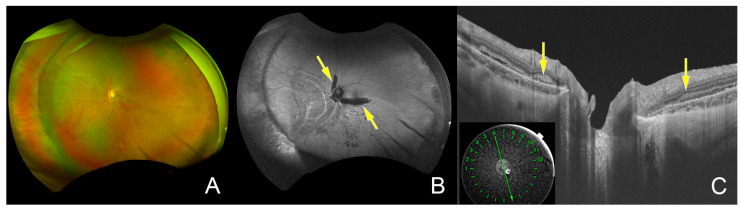
Postoperative widefield fundus photograph and FAF image. (**A**): The postoperative widefield fundus photograph shows that the atrophic retinal hole was located in the area indented by the buckle and the subretinal fluid was reduced. However, a subretinal hemorrhage was observed around the optic disc and the upper and lower arcade vessels. (**B**): Postoperative FAF image shows a hypo-FAF area (yellow arrows) corresponding to the subretinal hemorrhage, but no peripapillary starfish hypo-FAF was observed. (**C**): Postoperative optical coherence tomographic image showing hyperreflective subretinal hemorrhage (yellow arrows) near the optic disc.

**Table 1 jcm-12-01728-t001:** Ocular and systemic complications in Ehlers–Danlos syndrome with retinal detachment.

Authors, Year	Age	Sex	Subtype	Gene Mutation	Surgery	Ocular Complications	Systemic Complications
Beighton P 1970 [8]	36	woman	unknown	unknown	unknown	high myopia, strabismus, congenital glaucoma, microcornea, blue sclera	bilateral talipes equinovarus deformities of the feet, thoracolumbar kyphoscoliosis, frequent dislocations of both shoulders and subluxations, hiatus hernia, breast cystic masses
Bodanowitz S 1997 [9]	47	man	type VI	unknown	vitrectomy	blue sclera, keratoconus	hyperpigmentation and scarring of the skin, hyperextensibility of the thumb end links
Yasui H et al., 2003 [10]	23	man	musculocontractural type	CHST14	unknown	none	hyperextensible skin and hypermobile joint, anemia, hematoma
Malfait F et al., 2010 [11]	14 *	woman	musculocontractural type	CHST14	unknown	high myopia, glaucoma	large fontanel, flexion–adduction contractures of the thumb, kyphoscoliosis, skin fragility, hiatal hernia, craniofacial abnormalities
Malfait F et al., 2010 [11]	14 *	woman	musculocontractural type	CHST14	unknown	high myopia, glaucoma	muscular hypotonia, dislocations of the patellae, lumbar scoliosis, skin fragility
Janecke A et al., 2016 [12]	45	woman	musculocontractural type	CHST14	unknown	down slanting palpebral fissures, blue sclerae, microcornea, myopia, circumscribed retinal degenerations at the periphery	laxity of joints, frequent bruising, delayed wound healing, skin fragility with prolonged bleeding, dermal hyperelasticity
Whitlow S et al. 2020 [13]	33	woman	type VI	unknown	pneumatic retinopexy → vitrectomy and 360-degree scleral graft	high myopia	unknown
Lumi X et al., 2021 [14]	40	man	type IV	COL3A1	vitrectomy	high myopia	colon perforation, muscle and arterial rupture in both lower limbs, recurrent shoulder joint luxation
Current case	21	woman	musculocontractural type	dermatan sulfate epimerase	scleral buckling	high myopia, thin sclera	hyperextensible skin and hypermobile joint, prolonged bleeding, bradycardia

* = sisters.

## Data Availability

The data presented in this study are available upon request from the first author (Y.Y.).

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
