# Peer review of "Rhegmatogenous Retinal Detachment in Musculocontractural Ehlers–Danlos Syndrome Caused by Biallelic Loss-of-Function Variants of Gene for Dermatan Sulfate Epimerase"

_jcm, 2023, doi:10.3390/jcm12051728_

Round 1

Reviewer 1 Report

Ehlers–Danlos Syndrome (EDS) is a rare connective tissue disorder caused by biallelic loss-of-function variants in CHST14 (mcEDS-CHST14) or DSE (mcEDS-DSE), both of which are reported with ocular complications. The current manuscript reports a case with 24 years female EDS patient suffering from unilateral rhegmatogenous retinal detachment. Minimal surgeries including, scleral buckling, and retinal photocoagulation were performed. A postoperative widefield fundus photograph implies that RRD was successfully reattached. The manuscript was very well organized and very informative. 

Author Response

Answers for Reviewer 1:

We thank you and the reviewers for the comments of our manuscript.

Ehlers–Danlos Syndrome (EDS) is a rare connective tissue disorder caused by biallelic loss-of-function variants in CHST14 (mcEDS-CHST14) or DSE (mcEDS-DSE), both of which are reported with ocular complications. The current manuscript reports a case with 24 years female EDS patient suffering from unilateral rhegmatogenous retinal detachment. Minimal surgeries including, scleral buckling, and retinal photocoagulation were performed. A postoperative widefield fundus photograph implies that RRD was successfully reattached. The manuscript was very well organized and very informative. 

Answer: We thank you for your comments.

Reviewer 2 Report

The authors are reporting the management a case of rhegmatogenous detachment in a patient with a specific subtype of EDS. I believe the case should be published to expand the literature on RD in these rare cases. The manuscript is well written and my only comments are a minor review for infrequent language errors, and also I would like the authors to briefly describe the epidemiology of the disease and its subtypes (prevalence, incidence, sex/race predilection....etc.). 

Author Response

Answers for Reviewer 2

We thank you for the comments of our manuscript. The manuscript has been revised according to the comments, and our answers are included.

The authors are reporting the management a case of rhegmatogenous detachment in a patient with a specific subtype of EDS. I believe the case should be published to expand the literature on RD in these rare cases. The manuscript is well written and my only comments are a minor review for infrequent language errors, and also I would like the authors to briefly describe the epidemiology of the disease and its subtypes (prevalence, incidence, sex/race predilection....etc.). 

Answer: We thank you for your comments. The language errors have been revised throughout the manuscript. The prevalence, incidence of mcEDS-DSE has not been described because of an extremely rare disease. The following sentences were changed in line 49, “Eight mcEDS-DSE patients including 3 men and 5 women with two Indian, two Spanish, one Portuguese, one Pakistani, one Turkish, and one unknown have been described in the literature with ocular complications including blue sclera, strabismus, higher refractive error, and elevated intraocular pressure [3-5].”

In line 53, “These 8 patients with mcEDS-DSE have been reported to have craniofacial hypertelorism in 6 of 6 patients (100%), down-slanting palpebral fissures in 7 of 8 (88%), ear deformity in 7 of 8 (88%), congenital multiple contractures in 7 of 8 (88%), talipes equinovarus in 6 of 8 (75%), long or slender finger shape characteristics in 8 of 8 (100%), progressive foot deformities in 6 of 6 (100%), skin hyperextensibility in 3 of 7 (43%), fine or acrogeria-like palmar creases in 7 of 8 (88%), and large subcutaneous hematoma in 4 of 5 (80%) [5]. However, a complication of RD has not been reported. In contrast, 6 of the 41 patients with mcEDS-CHST14 have been reported to have RD [3].“

Reviewer 3 Report

In this well written case report Yoshikawa et at. present the first case of retinal detachment in patient with musculocontractual Ehlers-Danlos syndrome caused by biallelic loss-of-function variants of gene for dermatan sulfate epimerase. The authors provided a nice description of the etiology and characterestics of this phenotype with wide field imaging and explained the surgical managment.

minor concerns:

1- How can the authors explain the development of the postoperative subretinal hemorrhage? Could it be caused by the pressure of the buckle on the vortex veins over a thinned sclera? The authors should refer to other the possible reasons?

2- How was the retinal examination of the partner eye?

3- reformulate the following sentece: but showed the absence of intermediate optic media opacities of the cornea, anterior chamber, lens, and vitreous body  in line 69

4- Do the authors have OCT scans cutting the postoperative subretinal hemorrhage? if yes. please add one.

Author Response

Answers for Reviewer 3

We thank you for the comments of our manuscript. The manuscript has been revised according to the comments, and our answers are included.

In this well written case report Yoshikawa et al. present the first case of retinal detachment in a patient with musculocontractual Ehlers-Danlos syndrome caused by biallelic loss-of-function variants of gene for dermatan sulfate epimerase. The authors provided a nice description of the etiology and characteristics of this phenotype with widefield imaging and explained the surgical management.

minor concerns:

  1. How can the authors explain the development of the postoperative subretinal hemorrhage? Could it be caused by the pressure of the buckle on the vortex veins over a thinned sclera? The authors should refer to other the possible reasons?

Answer: We thank you for your comments. We believe that the compression of the vortex vein by the scleral buckle may have caused the exudative retinal detachment. The following sentence was changed in line 173,

“These findings suggested that the subretinal hemorrhage may have been from the peripapillary vessels due to scleral fragility and vascular fragility [18,19]. The placement of the scleral buckle may have stressed the area around the optic nerve where the shape of the eye changes the most causing the breakage of the peripapillary vessels leading to hemorrhage. The intraoperative surgical procedures or a rupture of the Bruch's membrane as is seen in simple hemorrhages in eyes with high myopia [20] did not appear to be the cause of the peripapillary subretinal hemorrhage according to the intraoperative and postoperative findings.”

  1. How was the retinal examination of the partner eye?

Answer: The following sentence was revised in line 89,

“The right eye appeared to be normal except for myopic changes of the fundus.”

  1. Reformulate the following sentence: but showed the absence of intermediate optic media opacities of the cornea, anterior chamber, lens, and vitreous body in line 69.

Answer: The sentence was revised in line 76 as follows.

“Slit-lamp examinations did not show a blue sclera (Figure 1), and the cornea, anterior chamber, lens, and vitreous body were clear.”

  1. Do the authors have OCT scans cutting the postoperative subretinal hemorrhage? if yes. please add one.

Answer: We have added a postoperative OCT scan in Figure 4C and the following sentence was inserted in line 133,

“C: Postoperative optical coherence tomographic image showing hyperreflective subretinal hemorrhage (yellow arrows) near the optic disc.”
